# Gingival Tissue MiRNA Expression Profiling and an Analysis of Periodontitis-Specific Circulating MiRNAs

**DOI:** 10.3390/ijms241511983

**Published:** 2023-07-26

**Authors:** Benita Buragaite-Staponkiene, Adomas Rovas, Alina Puriene, Kristina Snipaitiene, Egle Punceviciene, Arunas Rimkevicius, Irena Butrimiene, Sonata Jarmalaite

**Affiliations:** 1Institute of Biosciences, Life Sciences Centre, Sauletekio Ave. 7, LT-10257 Vilnius, Lithuania; benita.buragaite-staponkiene@gmc.vu.lt (B.B.-S.); kristina.snipaitiene@gmc.vu.lt (K.S.); sonata.jarmalaite@nvi.lt (S.J.); 2Institute of Odontology, Faculty of Medicine, Vilnius University, M. K. Ciurlionio St. 21, LT-03101 Vilnius, Lithuania; alina.puriene@mf.vu.lt (A.P.); arunas.rimkevicius@mf.vu.lt (A.R.); 3National Cancer Institute, Santariskiu St. 1, LT-08406 Vilnius, Lithuania; 4Clinic of Rheumatology, Orthopaedics Traumatology and Reconstructive Surgery, Institute of Clinical Medicine, Faculty of Medicine, Vilnius University, M. K. Ciurlionio St. 21, LT-03101 Vilnius, Lithuania; epunceviciene@santa.lt (E.P.); irena.butrimiene@santa.lt (I.B.); 5Centre of Rheumatology, Vilnius University Hospital Santaros Clinics, Santariskiu St. 2, LT-08410 Vilnius, Lithuania; 6Vilnius University Hospital Zalgiris Clinic, Zalgirio St. 117, LT-08217 Vilnius, Lithuania

**Keywords:** periodontitis, rheumatoid arthritis, miRNAs, biomarkers

## Abstract

This study aimed to identify the microRNAs (miRNAs) associated with periodontitis (PD) in gingival tissues, and to evaluate the levels of these selected miRNAs in the saliva and blood plasma among participants with and without rheumatoid arthritis (RA). A genome-wide miRNA expression analysis in 16 gingival tissue samples revealed 177 deregulated miRNAs. The validation of the miRNA profiling results in 80 gingival tissue samples revealed that the PD-affected tissues had a higher expression of miR-140-3p and -145-5p, while the levels of miR-125a-3p were significantly lower in inflamed tissues. After a thorough validation, four miRNAs, namely miR-140-3p, -145-5p, -146a-5p, and -195-5p, were selected for further analysis in a larger sample of salivary (*N* = 173) and blood plasma (*N* = 221) specimens. Increased salivary levels of miR-145-5p were associated with higher mean values of pocket probing depth and bleeding on probing index. The plasma-derived levels of miR-140-3p were higher among the participants with PD. In conclusion, the gingival levels of miR-140-3p, -145-5p, and -125a-3p were independently associated with PD presence and severity. The salivary and blood plasma levels of the target miRNAs were diversely related to PD. Similar miRNA associations with PD were observed among the participants with and without RA.

## 1. Introduction

MicroRNAs (miRNAs) are a group of small, single-stranded, non-coding RNA molecules directly involved in the regulation of gene expression through the post-transcriptional modulation of target messenger RNA molecules (mRNAs) [1]. MiRNAs cause a repression of the translation of mRNAs into proteins by the initiation of mRNA degradation or a repression of its translation [1]. MiRNAs are an integral part of various regulatory networks that fine-tune mRNA and, subsequently, protein expression levels, which leads to a downregulation or exacerbation of numerous cellular processes, such as carcinogenesis, immunity, and inflammation [2,3]. Aberrant miRNA expression has been associated with multiple diseases and conditions, providing promising therapeutic options for the precise management of diseases [4]. Moreover, disease-associated miRNA deregulation is reflected in various biological fluids, such as blood plasma and saliva, showing the utility of miRNAs for liquid-biopsy-based testing [4]. The prospect of miRNA utilization for diagnostic and therapeutic approaches has led to an ever-increasing number of studies analyzing the miRNA expression associations with various diseases, including periodontal pathology.

Periodontitis (PD) is a highly prevalent chronic inflammatory disease that is initiated by periodontal pathogenic bacteria accumulation on tooth surfaces [5]. A host’s immuno-inflammatory response to periodontal pathogens causes indirect damage to the periodontal tissues and consequently defines the clinical manifestation of PD [6]. The extent of the immune system’s involvement is associated with genetic predisposition, environmental factors, and various comorbidities, such as diabetes mellitus, rheumatoid arthritis (RA), and others [7]. The association between PD and RA has been widely documented, revealing similarities in the diseases’ pathogenesis, namely the overproduction of pro-inflammatory cytokines, connective tissue breakdown, and a chronic inflammatory state [8]. Moreover, significant correlations between the clinical statuses of PD and RA have been observed, while RA treatment with biologic disease-modifying antirheumatic drugs (bDMARDs) has been shown to affect inflammation in the periodontal tissues [9].

PD’s complex pathogenic pathways and various systemic interactions remain to be fully revealed. The discovery of miRNA involvement in the inflammation and modulation of immunity has provided additional insights into the etiopathogenesis of PD [3]. MiRNAs play crucial roles in periodontal disease pathophysiology, regulating cell differentiation, inflammation, and apoptosis. They are involved in the early stages and progression of PD and have been proposed as potential non-invasive biomarkers. Dysregulated miRNAs in saliva suggest they may serve as biomarkers for PD. Analyzing miRNA expression levels could aid with the early detection, diagnosis, and treatment of periodontal disease. However, significant pitfalls are to be avoided aiming to evaluate the miRNA associations with PD, as the number of identified human miRNAs continuously increases and the assessment of each miRNA’s role in PD pathogenesis may be challenging [10]. Therefore, the present study employed a multi-phase approach, which involved genome-wide miRNA profiling and potentially PD-associated miRNAs selection and validation in gingival tissues, with the assessment of the selected target miRNAs in the saliva and blood plasma among participants with and without RA.

## 2. Results

### 2.1. Participant Characteristics and Clinical Findings

Two-hundred and thirty participants were enrolled in the study and classified according to their periodontal and rheumatological status (see Section 4). The sociodemographic and anthropometric characteristics of the study participants are summarized in Appendix A, highlighting the observed significant differences between the PD+ and PD− groups regarding age, sex, tobacco usage, and oral-health-related behaviors.

The clinical outcome parameters of the periodontal and rheumatological examinations are presented in Table 1. Intragroup comparisons with regard to RA status revealed that the participants with both PD and RA had significantly higher values of mean clinical attachment loss (CAL), probing pocket depth (PPD), and number of missing teeth, as compared to PD patients without RA (*p* < 0.050).

### 2.2. Genome-Wide MiRNA Expression Profiling in Gingival Tissues

MiRNA expression profile was analyzed in 16 gingival tissue samples: PD-affected (PD+, *N* = 8) and PD-unaffected (PD−, *N* = 8). Both groups included four participants diagnosed with RA. Out of the 2569 available miRNAs, 760 were detected in ≥25% of the samples and were studied further (Figure 1A).

A hierarchical clustering analysis revealed the segregation of the samples into two distinct groups: PD+ and PD− (Figure 1B). A total of 177 differentially expressed miRNAs (fold change ≥ 1.5 or ≤ −1.5; *p* ≤ 0.050) were observed between the groups. In the PD-affected tissues, most miRNAs (*N* = 140, 79.1%) were upregulated as compared to the healthy tissues, including miR-30a-5p, -125a-3p, -126-5p, -140-3p, -145-5p, -146a-5p, -155-5p, -195-5p, -575, -630, -1273g-3p, and -3917 (Appendix A).

Considering the importance of RA in the modulation of the immuno-inflammatory response, a subgroup analysis was performed with regard to RA status. Firstly, only the tissues of participants without RA were assessed (PD+RA−, *N* = 4 vs. PD−RA−, *N* = 4). A total of 118 significantly deregulated miRNAs were revealed, with a trend towards miRNA upregulation (*N* = 77, 65.3%) in the PD-affected tissues. Secondly, the inflamed gingiva of participants with both diseases (PD+RA+, *N* = 4) were compared with the uninflamed gingival tissues of the healthy participants (PD−RA−, *N* = 4). Out of the 29 miRNAs with significantly altered expression, 22 were upregulated, including miR-140-3p, -550a-3-5p, and -765.

Based on the revealed associations, 15 potentially PD-related miRNAs were selected for further analysis, namely: miR-30a-5p, -125a-3p, -126-5p, -140-3p, -145-5p, -146a-5p, -155-5p, -195-5p, -423-5p, -550a-3-5p, -575, -630, -765, -1273g-3p, and -3917. In order to validate the detected changes, the selected miRNAs were further analyzed by performing quantitative reverse transcription PCR (RT-qPCR) in an extended cohort of gingival tissues (*N* = 80).

### 2.3. Selected MiRNA Expression Analysis in Gingival Tissue

Consistent with the microarray analysis, the tissues affected by PD had elevated levels of miR-140-3p (*p* = 0.013) and miR-145-5p (*p* ≤ 0.001), whereas the expression of miR-125a-3p (*p* = 0.001) was significantly lower in the inflamed tissues (Figure 2A).

A subgroup analysis of the non-arthritic individuals showed a similarly significant increased expression of miR-140-3p and miR-145-5p in the tissues affected by PD (Figure 2B). Meanwhile, the inflamed gingival tissues of the patients with RA had higher levels of miR-140-3p (*p* = 0.017) and close to significantly higher levels of miR-195-5p (*p* = 0.070), while lower levels of miR-125a-3p (*p* = 0.015) were observed compared to healthy participants (Figure 2C). After adjusting for age and sex, a multivariate analysis revealed that increased odds of having PD were associated with an increased gingival expression of miR-140-3p (OR = 2.174; 95% CI = 1.020 to 4.636, *p* = 0.044) and -145-5p (OR = 2.124; 95% CI = 1.168 to 3.559, *p* = 0.004), and decreased expression of miR-125a-3p (OR = 1.699; 95% CI = 1.158 to 2.494, *p* = 0.007).

The MiRNA expression in gingival tissues was moderately correlated with the clinical status of PD (Figure 3).

An assessment of PD severity defined by stages revealed that higher gingival levels of miR-140-3p (*p* = 0.003) and -145-5p (*p* < 0.001) and lower levels of miR-125a-3p (*p* = 0.007) were observed among the participants with severe PD (Stage IV and Stage III), as compared to healthy participants. The gingival tissues of patients with moderate PD (Stage II) had higher levels of miR-195-5p (*p* = 0.039) and lower levels of miR-125a-3p (*p* = 0.014).

Significant associations of the target miRNAs were observed with the mean values of the periodontal outcome variables, including the CAL, PPD, bleeding on probing (BOP), and bone loss (BL). The participants with pronounced gingival bleeding on probing (mean BOP ≥ 33) had higher gingival levels of miR-140-3p (*p* = 0.009) and miR-145-5p (*p* < 0.001) and lower levels of miR-125a-3p (*p* = 0.002). Similar associations that were observed for the other assessed periodontal outcome parameters are presented in Figure 4.

As illustrated in Figure 3, the RA status, represented by the score of DAS28-CRP and the use of disease-modifying antirheumatic drugs DMARDs, was found to be moderately associated with the expression of gingival miRNA. A univariate analysis revealed that individuals receiving biologic DMARDs (bDMARDs) had lower levels of miR-140-3p and miR-195-5p (*p* = 0.004 and *p* = 0.036, respectively) and higher levels of miR-125a-3p (*p* = 0.017), compared to those who were not receiving bDMARDs.

Following a genome-wide miRNA expression analysis in the gingival tissues, miRNA selection was performed for further investigation in bodily fluids, with 173 saliva and 221 plasma samples.

MiR-140-3p, miR-145-5p, and miR-195-5p were selected for further analysis based on their observed increased expressions related to PD presence and/or severity, whilst miR-146a-5p was selected due to its previously described significant associations with PD, as well as the importance of miR-146a-5p in the inflammation modulation in PD pathogenesis [11].

### 2.4. MiRNA Analysis in Saliva

The salivary levels of miR-140-3p, -145-5p, -146a-5p, and -195-5p were not significantly different with regard to PD and/or RA presence, however, significant associations were revealed after assessing the salivary miRNA associations with the clinical–pathological parameters, including PPD and BOP. In line with the gingival tissue analysis, the level of circulating miR-145-5p was 1.5-fold higher among the participants with a mean PPD of ≥3 mm, as compared to those with a PPD of <3 mm (*p* = 0.034). Similarly, a mean BOP value of ≥33% was associated with an increase in miR-145-5p salivary levels (*p* = 0.042), as compared to participants with lower BOP values. An analysis of PD severity defined by stages revealed that participants with stage IV PD had lower salivary levels of miR-195-5p, as compared to participants with less advanced disease (*p* = 0.039).

### 2.5. MiRNA Analysis in Blood Plasma

A plasma-derived miRNA assessment revealed that patients with PD had higher plasma levels of miR-140-3p (*p* = 0.030), as compared to participants without PD. A subgroup analysis among the participants without RA revealed that PD was associated with higher plasma levels of miR-145-5p (*p* = 0.020) and lower levels of miR-195-5p (*p* = 0.030), as compared to periodontally healthy participants.

Significant alterations in the plasma miRNA levels were observed with regard to PD severity. It was identified that severe PD (Stage III and IV) was associated with an increase in miR-195-5p (*p* = 0.007) and a decrease in miR-145-5p plasma levels (*p* < 0.001), as compared to participants with less advanced PD. In a similar manner, the miRNAs showed a different abundance in relation to PD clinical variables. Amongst the participants with a mean CAL value of ≥2.5 mm, a significant increase in the level of miR-146a-5p (*p* = 0.026) and a decrease in miR-145-5p (*p* = 0.002) levels were identified (Figure 4A). An analysis regarding PPD revealed a significantly higher level of miR-146a-5p (*p* = 0.045) and lower level of miR-145-5p (*p* = 0.002) in a group of patients with a PPD of ≥ 3 mm (Figure 4B).

Similar differences were identified when the associations with BL and BOP were analyzed (Appendix A).

### 2.6. Comparison of Bodily Fluids-Circulating MiRNAs

The relative quantities of miR-140-3p, miR-145-5p, -146a-5p, and -195-5p significantly differed between the gingival tissue, saliva, and blood plasma collected from the same individuals (Figure 5). A large variation in the miRNA levels was observed between the salivary- and plasma-derived miRNAs: the amount of miR-145-5p was 25.6-fold higher in the saliva, while miR-195-5p was 12.4-fold lower in the saliva as compared to the plasma (*p* < 0.001). In the gingival tissues, the expressions of miR-146a-5p (*p* < 0.001) and miR-195-5p (*p* < 0.001) were higher and that of miR-145-5p (*p* < 0.001) was lower than those detected in the saliva.

### 2.7. Diagnostic Performance of Target MiRNAs

The diagnostic potential of the selected miRNAs was assessed by performing a receiver operating characteristic (ROC) analysis with an area under curve (AUC) value calculation. In the gingival tissue, significant AUC values were observed for miR-125a-3p (AUC = 0.71; 95% CI = 0.59 to 0.83, *p* = 0.002), miR-140-3p (AUC = 0.70; 95% CI = 0.57 to 0.82, *p* = 0.003), and miR-145-5p (AUC = 0.74; 95% CI = 0.62 to 0.85, *p* < 0.001). A combination of the aforementioned miRNAs had an improved AUC value of 0.83, as shown in Figure 6A.

The salivary levels of the target miRNAs failed to discriminate between the PD+ and PD− groups. Meanwhile, in the blood plasma, neither of the separate biomarkers were able to separate between the ill and healthy cases; however, a combination of all the miRNAs showed a weak but statistically significant AUC value (AUC = 0.58; 95% CI = 0.51 to 0.66, *p* = 0.036) (Figure 6B).

## 3. Discussion

In the present multiphase study, we employed a step-by-step approach in order to select the miRNAs associated with periodontal disease, and evaluated the levels of these miRNAs in saliva and blood plasma. Regarding the fact that periodontal disease is a condition largely influenced by a host’s immuno-inflammatory response to periodontal pathogens, we included a number of participants with a systemic condition that is known to modify the immune system’s functioning—RA [8]. The participants with RA were regarded as a subgroup and were utilized to emphasize the differences in a PD+ vs. PD− analysis among the patients with and/or without RA.

The genome-wide miRNA expression profiles were analyzed in the gingival tissues via a high-throughput analysis. It was observed that there were significant differences in the miRNA expression profiles between the PD-affected and healthy gingiva, with the majority of the 177 significantly deregulated miRNAs being upregulated in inflamed gingival tissues. This suggests that miRNAs are actively involved in the inflammatory modulation responsible for the development of PD. Microarray-based miRNA profiling has been used in several studies to compare healthy and inflamed gingival tissues, varying largely in terms of sample size and the number of deregulated miRNAs [12]. Additionally, a recent systematic review found that inconsistent results between these studies could be linked to the definition of periodontal disease, since many studies were conducted prior to the implementation of the current American Academy of Periodontology (AAP) and the European Federation of Periodontology (EFP) classification [13]. To enhance the precision of identifying the miRNAs associated with PD, we utilized a reliable sample of 16 tissues in our study and conducted a subgroup analysis with regard to RA status. This approach led us to identify 15 miRNAs that may potentially be associated with PD, including miR-30a-5p, -125a-3p, -126-5p, -140-3p, -145-5p, -146a-5p, -155-5p, -195-5p, -423-5p, -550a-3-5p, -575, -630, -765, -1273g-3p, and -3917.

In the resulting validation stage of the study, we conducted an analysis of the miRNAs in the gingival tissue by means of RT-qPCR and found that miR-140-3p and miR-145-5p were significantly overexpressed in the PD-affected tissues. The upregulation of these miRNAs was found to be associated with both the presence of PD and the clinical status of PD. Our results also demonstrated that higher average measures of CAL, PPD, BL, and BOP were linked to increased levels of miR-140-3p and miR-145-5p. These findings are consistent with previous research showing that miR-140-3p is upregulated in PD-affected tissues compared to healthy gingiva [14]. It is suggested that miR-140-3p may play a role in bone remodeling and osteogenic differentiation [15]. Liu et al. showed that miR-140-3p regulates the osteogenic differentiation of bone marrow mesenchymal stem cells (BMSCs) through *Spred2* [16]. Moreover, it was also demonstrated that miR-140-3p directly targets lysine methyltransferase 5B (*Kmt5b*), and its depletion might reverse the defective osteo/odontogenic differentiation of dental pulp stem cells under hypoxic conditions [17]. This miRNA has been shown to play a role in regulating inflammation in various inflammatory model systems such as RA [18], osteoarthritis [19], and acute lung injury [20]. It is suggested that miR-140-3p suppresses the production of pro-inflammatory cytokines and reduces the inflammatory response.

Similarly, miR-145-5p has been found to have a negative impact on bone formation, with studies showing that inhibiting this miRNA can increase osteogenic differentiation [21]. Additionally, previous research has linked higher levels of tumor necrosis factor-alpha (TNF-α) to an increased expression of miR-145-5p [22]. MiR-145-5p targets *Bach2* to mediate TNF-induced apoptosis in human gingival epithelial cells [21]. Moreover, miR-145-5p regulates the proliferation and chondrogenic differentiation of BMSCs through targeting the *Smad4* pathway in osteoarthritis [23]. MiR-145-5p has been shown to have anti-inflammatory effects in atherosclerosis [24] and inflammatory bowel disease [25], where it targets the genes involved in the inflammatory responses and inhibits the production of pro-inflammatory cytokines. However, it was demonstrated that, in asthma, miR-145-5p may have a pro-inflammatory effect [26].

The present study also revealed significant miR-125a-3p downregulation in the PD-affected tissues, which moderately correlated with the PD outcome parameters. Contrary to miR-140-3p and miR-145-5p, increased levels of miR-125a-3p have a positive effect on bone remodulation, meanwhile the inhibition of miR-125a-3p promotes osteoclastogenesis [27]. The expression levels of miR-195-5p were significantly increased in the gingival tissues of the individuals with moderate PD. MiR-195-5p has been linked to osteogenic differentiation via *Wnt3a*. [28,29]. This miRNA has demonstrated anti-inflammatory properties in inflammatory models such as atherosclerosis [30], while it has also been shown to have pro-inflammatory effects in osteoarthritis [31] and ulcerative colitis [32], whereas hsa-miR-146a-5p may promote differentiation in dental stem cells by targeting *Traf6* via the NF-κB signaling pathway [33]. Another study discovered that miR-146a-5p inhibited IL-6 and IL-1 production via the IRAK1/TRAF6 pathway, and thereby suppressed the IL-1-induced inflammatory factors in cementoblast-derived cells in vitro [34]. MiR-146a-5p is a key negative regulator of the innate immune response. It has been shown to reduce inflammation in RA [35] and multiple sclerosis [36].

A careful consideration of the involvement of miRNAs in immunity and inflammation, as well as an analysis of the RT-qPCR data, led to the identification of four miRNAs (miR-140-3p, -145-5p, -146a-5p, and -195-5p), which were selected for further assessments in both the saliva and blood plasma.

In accordance with the gingival tissue analysis, the salivary levels of miR-145-5p were associated with the clinical status of PD, however, a difference was only observable when the periodontal outcome variables were assessed, namely PPD and BOP. Despite promising results in salivary miRNA-based diagnostic approaches for oral malignancies [37], the salivary miRNA associations with PD have been proven to be less consistent based on recent findings. According to the findings of a pilot study carried out by Han et al., salivary miRNAs extracted from extracellular vesicles (EV) are more effective for diagnostic purposes compared to miRNAs extracted from whole saliva [38]. An analysis of the whole-saliva miRNAs did not show any association between the salivary levels of miR-140-3p and miR-146a-5p with the presence of PD, however, these miRNAs were found to be significantly upregulated in an EV-based miRNA analysis. In a separate study, an evaluation of the levels of 84 miRNAs in the saliva revealed that only the levels of miR-381-3p were significantly elevated in individuals with severe PD [39]. Although a link was found in the current study between PD and the levels of the miRNAs in saliva, it is worth noting that the reference data utilized were derived from participants with advanced stages of PD. Ideally, diagnostics based on salivary miRNAs should facilitate the detection of early PD or susceptibility to the disease. Therefore, alternative techniques for assessing the miRNAs in saliva, other than a whole-saliva miRNA analysis, may be beneficial for PD evaluation.

The present study found that changes in the levels of the target miRNAs in the plasma were moderately associated with both the presence and severity of PD, as determined by PD stages and the periodontal outcome parameters. The plasma levels of miR-140-3p were elevated in the PD-affected participants, which corresponded with the increase in miR-140-3p expression in the gingival tissues. In contrast, the plasma levels of miR-145-5p were lower, despite higher levels of miR-145-5p expression in the gingival tissues. Similar to this study, a recent investigation of plasma and saliva EV miRNA levels also observed differing levels of miRNA expression in different samples [40]. Previously, several studies have indicated a correlation between blood plasma miRNAs and the status of periodontal disease [12]. While the utility of plasma-miRNA-based diagnostic tests for periodontal disease remains uncertain, the consistent link between PD and changes in the plasma miRNA levels observed in this and other studies contributes to a better understanding of the relationship between PD and overall systemic health.

A noteworthy revelation of the present study was the observed variance in miRNA levels with regard to RA status. Despite the fact that, in general, the levels of the assessed target miRNAs were in accordance among the participants with and without RA, a variation in the FC values and level of significance was observed. Furthermore, the use of bDMARDs was found to have a slight but significant effect on gingival miRNA expression.

Previously, it has been shown that bDMARDs may lower the inflammatory cytokine levels in gingival tissues [41]. Despite controversy regarding the significance of bDMARDs’ impact on the overall clinical status of PD, [42] it is apparent that biologic medication is related to changes in gingival miRNA levels. Until recently, the majority of studies assessing miRNA associations with PD enrolled systemically healthy participants [12]. Several studies have evaluated the influence of systemic conditions on PD-associated miRNA levels, such as diabetes mellitus, acute coronary syndrome, and obesity [43,44,45]. In accordance with the present study, alterations in miRNA levels were observed with regard to systemic health, implying that it is desirable to enroll participants with systemic conditions, aiming to fully evaluate the miRNA associations with PD in future studies.

Although the study employed a rigorous methodology for selecting the miRNAs associated with PD and had a considerable sample size, it was not immune to certain limitations. In particular, there were noteworthy demographic disparities between the PD+ and PD- groups, in terms of the participant age and sex distribution. It is notable that PD exhibits a higher prevalence in older age groups, and emerging evidence has demonstrated a correlation between alterations in miRNA expression within aging cohorts and their association with the regulation of inflammatory pathways and the cellular senescence in periodontal ligament cells [46]. In the present study, to minimize the impact of age and sex, we performed a multivariate analysis with adjustment for the aforementioned variables. Another limitation of the study was the difference in the saliva and plasma sample sizes, which we describe in detail in Section 4.

The results of this study indicate that miRNAs are actively involved in the inflammatory modulation responsible for the development of PD and that these miRNAs can potentially be utilized as diagnostic markers for the disease. However, further research is needed to fully evaluate the miRNA associations with PD, especially in the context of systemic conditions. While the current study sheds light on certain aspects of the topic, it may also serve as a foundational work for further, more targeted research in the field.

## 4. Materials and Methods

### 4.1. Study Population and Clinical Examination

In total, 230 participants were enrolled in the cross-sectional study. Individuals receiving dental or rheumatological treatment at the Vilnius University Hospital (VU) Santaros and Zalgirio Clinics between 2018 and 2020 were invited to participate in the study, subject to specific inclusion and exclusion criteria. The research details were thoroughly presented to the participants and written informed consent was obtained prior to their enrollment. The study protocol was approved by the Vilnius regional biomedical research ethics committee (No. 158200-18-992-500). The experiments were carried out according to the principles of the Declaration of Helsinki of 1975, revised in 2013.

A periodontal assessment was conducted by a single researcher who is a certified periodontist (A.R.). The assessment comprised a clinical examination, X-ray analysis, and questionnaire. The periodontal examination involved full-mouth probing at six pre-defined sites per tooth, namely the mesio-buccal, mid-buccal, disto-buccal, mesio-lingual, mid-lingual, and disto-lingual sites, utilizing a specialized periodontal probe (PCPUNC 15; Hu-Friedy, Chicago, IL, USA). The intra-examiner reliability of the periodontal examination was assessed by the means of Kappa coefficient, achieving a value of 0.86 for CAL with difference of ±1 mm.

During the periodontal assessment, the following parameters pertaining to periodontal outcome were recorded: CAL, PPD, BOP, and number of missing teeth. To identify the potential bone loss (BL), the panoramic X-ray scans obtained during the clinical examination were meticulously scrutinized via a computerized analysis, which was carried out with strict adherence to an established protocol described previously [28]. A questionnaire was administered to evaluate the patients’ sociodemographic characteristics and oral-health-related habits.

The periodontal diagnosis was determined based on the 2018 AAP/EFP classification of the periodontal diseases guidelines [47]. As per these guidelines, a diagnosis of periodontitis was established if there was clinical evidence of PD-related interdental CAL at ≥2 non-adjacent teeth or a buccal or oral CAL of ≥3 mm with pocketing of >3 mm detectable at ≥2 teeth. The severity of the periodontitis was determined according to staging criteria, with Stage I indicating an initial phase of the disease, Stage II signifying moderate PD, Stage III—severe periodontitis, and Stage IV—advanced severe periodontitis.

A subgroup of participants with diagnosed RA was formed from patients undergoing rheumatological treatment at the VU Hospital Santaros Clinics. In total, 91 participants diagnosed with RA, who met the ACR/EULAR 2010 rheumatoid arthritis classification criteria, were enrolled in the study and were referred for periodontal status evaluation [29]. A single, qualified researcher rheumatologist (E.P.) conducted the assessment of RA status. The clinical status of the RA was evaluated by a Disease Activity Score 28 C-reactive protein (DAS28-CRP), as well as patient-derived questionnaires provided to all the RA subjects, namely the Rheumatoid Arthritis Impact of Disease (RAID) and the Health Assessment Questionnaire (HAQ) [48,49,50]. The RAID tool was utilized to measure the extent of the impact of the disease, whereas the HAQ assessed their disability in performing daily life activities. The types of RA treatments employed, such as glucocorticoids or conventional synthetic or biologic disease-modifying antirheumatic drugs, were duly documented.

During the recruitment period, any participant aged 18 years or older was considered as eligible for inclusion if no exclusion criteria were present. The participant exclusion criteria were full edentulism, periodontal treatment within 6 months, oral cavity cancer, and any known systemic modifier of periodontal status: autoimmune disorders (excluding RA), endocrine disorders (i.e., diabetes mellitus), and pharmaceutical therapy known to affect periodontal status. Data regarding comorbidities and pharmaceutical therapy were obtained from medical records.

### 4.2. Sample Collection

The gingival tissue sampling for the participants with PD was performed from the deepest periodontal pocket (PPD ≥ 5 mm) during initial periodontal treatment. Diseased gingival tissues presented with clinical signs of inflammation—erythema and BOP. Healthy gingival tissues were procured from patients undergoing non-research-related interventional procedures, such as surgical crown lengthening. A sample was considered to be healthy gingiva if no clinical signs of inflammation were observed, PPD was less than 3 mm, and no BL and CAL were detected. The healthy samples were collected from the participants without PD. Following collection, the gingival tissue samples were promptly submerged into an aqueous RNA stabilization solution RNAlater™ (Thermo Fisher Scientific (TFS), Waltham, MA, USA).

Unstimulated whole saliva was collected following a modified protocol reported previously [51]. The participants were instructed to abstain from food and drink for a minimum of 2 h before undergoing the saliva sampling. Prior to collection, the participants were asked to rinse their mouth with distilled water for 1 min. About 2 mL of saliva was aspirated from each participant into a sterile tube and subsequently subjected to centrifugation at 2600× *g* for 15 min. A salivary supernatant was carefully separated from the cellular phase and transferred into separate, sterile 2 mL tubes. To protect against potential RNA degradation, the RNase inhibitor SUPERase™ (TFS, Waltham, MA, USA) was added into each collected saliva sample.

Blood sampling was obtained by collecting 10 mL of peripheral blood using sterile BD Vacutainer^®^ tubes (Becton-Dickinson, Barricor™, Franklin Lakes, NJ, USA) containing dipotassium ethylenediaminetetraacetic acid (K2 EDTA) anticoagulant. The samples were then centrifuged at 2000× *g* for 15 min, allowing for the separation of the plasma from the cellular fraction. The extracted plasma was subsequently transferred into sterile 2 mL tubes for later analysis.

All the samples were collected by the same specialist (Adomas Rovas), who performed the periodontal examinations.

### 4.3. RNA Extraction

The total RNA was extracted from snap-frozen, ground gingival tissue and liquid biopsy samples of the saliva and blood plasma, using Trizol™ (Invitrogen, TFS, USA) reagent and an miRNeasy Mini Kit (Qiagen, Hilden, Germany), respectively, according to the manufacturers’ instructions. Following the addition of Trizol™ reagent to the gingival tissue lysate, chloroform was introduced, and the mixture was centrifuged. The RNA was subsequently isolated from the aqueous phase of the mixture by precipitating with isopropanol, followed by washing with ethanol and air-drying, before being dissolved in RNase-free water. For the liquid biopsy samples treated with QIAzol^®^ Lysis Reagent (Qiagen, Hilden, Germany), 25 fmol of synthetic spike-in control cel-miR-39 (manufacturer’s assay ID—000200, Applied Biosystems (ABI), TFS, Waltham, MA, USA) and chloroform were added. The RNA purification followed the manufacturer’s instructions, with the purified RNA being eluted in RNase-free water. The quality and quantity of the extracted RNA were assessed using a NanoDrop 2000 spectrophotometer (Thermo Scientific, Waltham, MA, USA) and Qubit 4 fluorometer (Invitrogen, TFS, Waltham, MA, USA), and the extracted RNA was stored at a temperature of −80 °C until use.

### 4.4. MiRNA Microarray Analysis

The global miRNA expression profiling was conducted using a total of 16 gingival tissue samples. The clinic-pathological data of the subjects examined at this stage are presented in Appendix A. Briefly, a quantity of 100 ng of total RNA was subjected to labeling with 3-pCp cyanine dye using miRNA Complete Labeling and Hyb Kit, along with a MicroRNA Spike-In Kit (both from Agilent Technologies (AT), Santa Clara, CA, USA). The labeled samples were then hybridized onto Human microRNA 8 × 60K format microarrays (AT) at 55 °C for 20 h. Following hybridization, the microarray slides were scanned using an Agilent SureScan scanner (AT). An analysis of the array images, along with further data processing, was conducted using the Feature Extraction v10.7 software packages (AT).

The microarray data were analyzed using GeneSpring GX v.14.9 software (AT). The data were 80th percentile normalized and the miRNAs that were under-expressed in the periodontal tissues and/or expressed in <25% of the samples were removed from the further analysis. For the comparison of the two groups of interest, a fold change (FC) was calculated and the unpaired *t*-test was applied. The subsequent analysis was restricted solely to the subset of miRNAs with an FC of ≥1.5 (*p* ≤ 0.050).

### 4.5. TaqMan Low Density Arrays

To assess the expression levels of the specific miRNAs in the 80 gingival tissue samples, custom-designed TaqMan Low Density Arrays (ABI, TFS) were used. The RNA was reverse transcribed into cDNA using the TaqMan MicroRNA Reverse Transcription Kit and specific RT Primer Pool mix (ABI, TFS), following the manufacturer’s protocol for cDNA synthesis without preamplification. The reverse transcription (RT) reaction was performed in 15 µL of reaction volume and contained 350 ng of input RNA. For the RT-qPCR, a reaction mix containing 2× TaqMan Universal Master Mix II, No UNG, nuclease-free water, and synthesized cDNA was loaded onto TLDA cards, which were then analyzed using the ViiA7 Real-Time PCR System and ViiA 7 Software v1.2 (all from ABI, TFS).

### 4.6. CDNA Synthesis and Quantitative Reverse Transcription Polymerase Chain Reaction

The levels of miR-140-3p, -145-5p, -146a-5p, and -195-5p were quantified in both the blood plasma and saliva samples using TaqMan^®^ MicroRNA Assays (manufacturer’s assay IDs—002234, 002278, 000468, and 000494, respectively, ABI, TFS) by means of RT-qPCR. RT was performed using the TaqMan^®^ MicroRNA Reverse Transcription Kit (ABI, TFS), following the manufacturer’s instructions. The RT reaction volume for each gene in each sample was 7.5 μL, containing specific stem-loop RT primers and 2.5 μL (1–10 ng) of total RNA. For the RT-qPCR, 1.33 μL of RT product was added to 2× TaqMan™ Universal Master Mix II, no UNG, 20× miRNA assays (both from ABI, TFS), and RNase-free water, resulting in a final reaction volume of 10 μL. All the samples were analyzed in technical triplicates using the ViiA7 Real-Time PCR System and ViiA 7 Software v1.2 (both from ABI, TFS).

### 4.7. Sample Size Calculation and Study Design

The sample size calculation was performed regarding our previously published data on miRNA expression in gingival tissue [52]. A minimum of 80 participants was necessary to reveal a medium difference (effect size of 0.6), with a test power of 85% and alpha error probability of 0.050.

In total, 240 participants agreed to participate and underwent clinical examination and sample collection. Following an insufficient initial blood sampling procedure, a number of participants (*N* = 19) did not continue to collaborate. Meanwhile, only saliva samples with a sufficient RNA quantity were included in the data analysis (*N* = 173). Regarding the fact that the study was primarily focused on an miRNA analysis, only good-quality biological samples were analyzed, resulting in final cohort of 230 participants. An overview of the study design is presented in Figure 7.

For the data analysis, all the study participants were allocated into two groups according to PD status: participants with diagnosed PD (PD+ group) and periodontally healthy participants (PD− group). Both the PD+ and PD− groups included a number of participants with diagnosed RA; therefore, when appropriate, the subgroups with regard to RA status were presented as follows: PD+RA+, PD+RA−, PD−RA+, and PD−RA− (Table 1).

### 4.8. Statistical Analysis

The continuous variables were presented as mean ± standard deviation (SD). To compare the clinical variables, a Shapiro–Wilk normality test was conducted, followed by a Student’s *t*-test or a Mann–Whitney U test. For the categorical variables, the chi-squared test was used. The MiRNA levels underwent normalization using the spiked-in cel-miR-39 and were subsequently log2-transformed. The correlation between the gingival miRNA expression and clinical variables was analyzed by using a Spearman correlation analysis. The MiRNA multivariate associations were analyzed by performing logistic regression and calculating the odds ratio (OR) with 95% confidence intervals (CI). The diagnostic performance of identifying PD patients from periodontally healthy individuals was evaluated by determining the area under the receiver-operator characteristic (ROC) curve (AUC). A significance level of *p* ≤ 0.050 was considered to be statistically significant.

Statistical software programs for the sample size calculation and data analysis were employed, including GPower 3.1.9 (University of Düsseldorf, Germany), GenEx v7.0 (MultiD Analyses AB, Göteborg, Sweden), and SPSS 20.0 (IBM, Armonk, NY, USA).

## 5. Conclusions

A high-throughput analysis of gingival tissues revealed significant variance in the miRNA expression profiles in PD-affected tissues as compared to healthy gingiva. Higher expressions of miR-140-3p and miR-145-5p and a lower expression of miR-125a-3p in the gingival tissues was associated with PD presence and PD severity. Increased salivary miR-145-5p levels were related to higher mean values of PPD and BOP. The levels of plasma-derived miRNAs were inconsistently associated with PD presence and severity. Similar miRNA associations with PD were observed among the participants with and without RA, However, the use of bDMARDs revealed altered miRNA expression patterns in the gingival tissues. The present study presented an overview, which highlights the aspects to be considered in future studies aiming to assess the miRNA associations with PD.

## Figures and Tables

**Figure 1 ijms-24-11983-f001:**
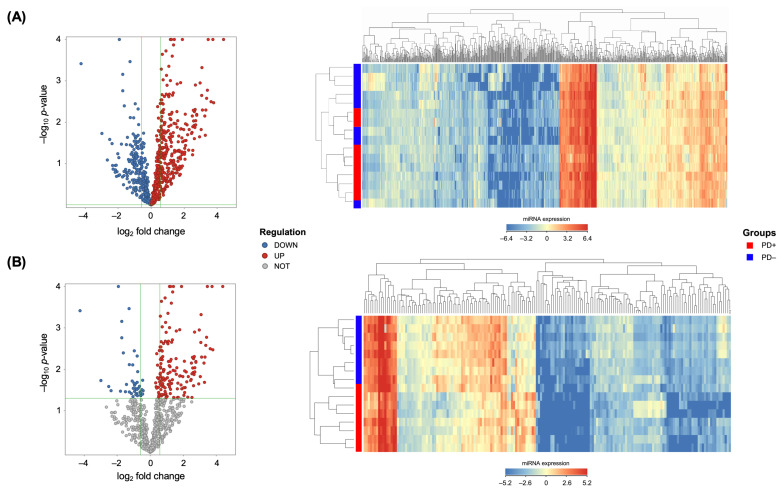
Volcano plot and hierarchical clustering of differentially expressed miRNAs in periodontitis-affected (PD+) vs. control group (PD−) gingival tissue samples: (**A**) miRNAs that were detected in ≥25% of the samples (*N* = 760) and (**B**) only significantly (*p* ≤ 0.050) deregulated miRNAs (*N* = 177).

**Figure 2 ijms-24-11983-f002:**
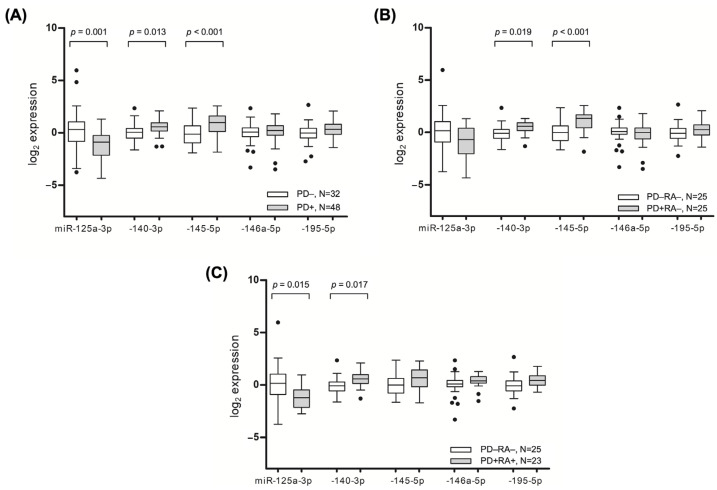
Relative expression of miR-125a-3p, -140-3p, -145-5p, -146a-5p, and -195-5p in gingival tissues collected from patients suffering from periodontitis (PD), rheumatoid arthritis (RA), and healthy participants. (**A**) Comparison of miRNA expression between PD+ and PD− groups, regardless of RA status. (**B**,**C**) Subgroup analysis comparing tissue samples from PD patients with and without RA (PD+RA+ and PD+RA−, respectively), as well as healthy participants (PD−RA−). The bands inside the boxes indicate the median, the whiskers show the data interval, and the dots represent outliers.

**Figure 3 ijms-24-11983-f003:**
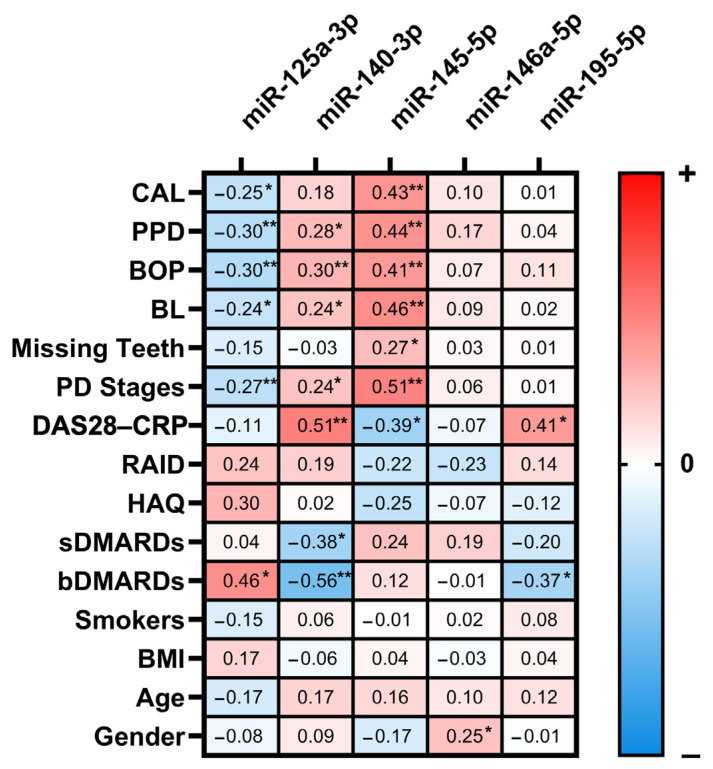
Correlation between gingival miR-125a-3p, 140-3p, -145-5p, -146a-5p, and -195-5p expression and clinical variables. The Spearman correlation coefficients are color-coded according to the scalebar, with red signifying a positive correlation, blue indicating a negative correlation, and white no correlation. The results showing significant correlations are indicated as follows: * *p* ≤ 0.050, ** *p* ≤ 0.010. bDMARDs: biologic disease-modifying antirheumatic drugs; BL: bone loss; BMI: body mass index; BOP: bleeding on probing; CAL: clinical attachment loss; DAS28-CRP: Disease Activity Score 28-joint count C reactive protein; HAQ: health assessment questionnaire; PD: periodontitis; PPD: periodontal probing depth; RAID: rheumatoid arthritis impact of disease; and sDMARDs: synthetic disease-modifying antirheumatic drugs.

**Figure 4 ijms-24-11983-f004:**
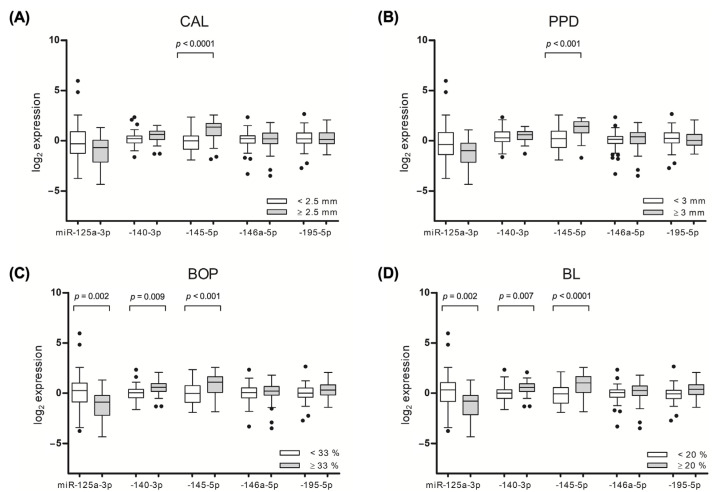
Relative expression of miR-125a-3p, -140-3p, -145-5p, -146a-5p, and -195-5p in gingival tissue evaluated by means of RT-qPCR. Comparison of cut-off points of (**A**) mean clinical attachment loss (CAL); (**B**) periodontal probing depth (PPD); (**C**) bleeding on probing (BOP); and (**D**) bone loss (BL). The bands inside the boxes indicate the median, the whiskers show the data interval, and the dots represent outliers.

**Figure 5 ijms-24-11983-f005:**
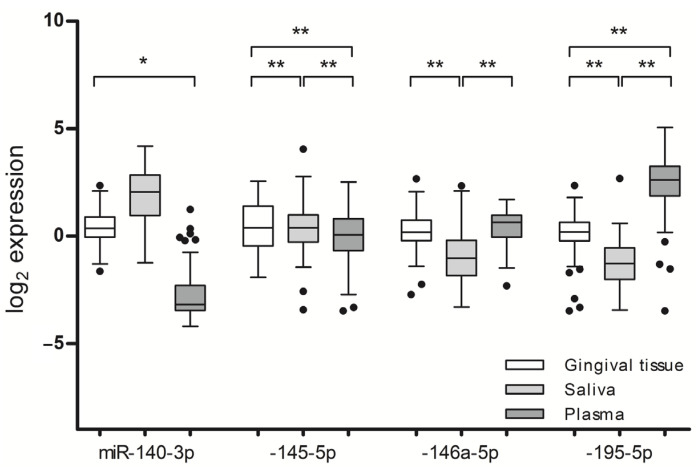
Relative quantities of miR-140a-3p, -145-5p, -146a-5p, and -195-5p in gingival tissue, saliva, and blood plasma collected from corresponding participants. * *p* ≤ 0.050 and ** *p* ≤ 0.010. The bands inside the boxes indicate the median, the whiskers show the data interval, and the dots represent outliers.

**Figure 6 ijms-24-11983-f006:**
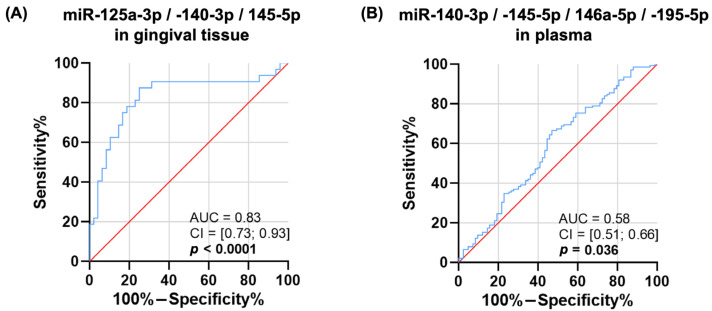
Receiver operating characteristic curves analysis identifying periodontitis patients from periodontally healthy participants. Area under the curve (AUC) for (**A**) combination of gingival miR-125a-3p, -140-3p, and -145-5p expression; (**B**) combination of plasma miR-140-3p, -145-5p, -146a-5p, and -195-5p levels.

**Figure 7 ijms-24-11983-f007:**
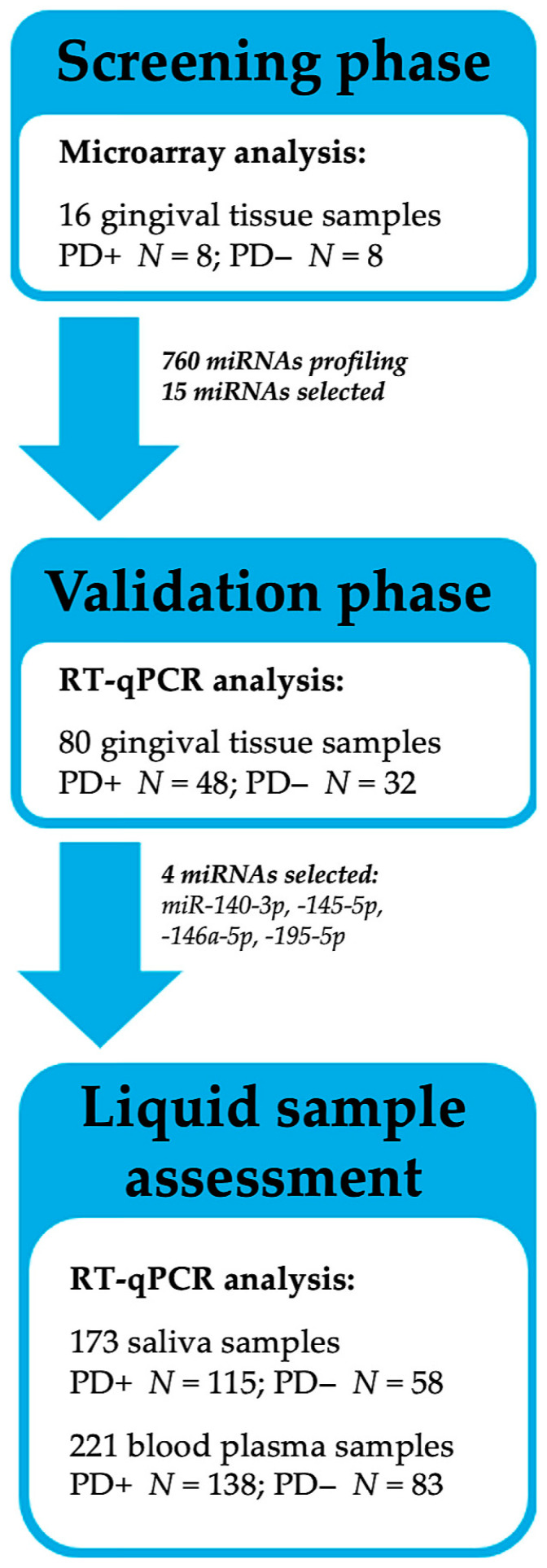
An overview of study design.

**Table 1 ijms-24-11983-t001:** Periodontal and rheumatological status (*N* or mean ± SD) of study participants.

Periodontal Status, *N*	PD+, 144		PD−, 86	
RA status, *N*	RA+, 61	RA−, 83	RA+, 30	RA−, 56
**PD clinical parameters**				
CAL (mm)	2.32 ± 0.89 *	2 ± 0.94 *	1.31 ± 0.14	0.98 ± 0.66
	2.13 ± 0.93 ***		1.09 ± 0.71 ***	
PPD (mm)	2.82 ± 0.58 **	2.57 ± 0.49 **	2.06 ± 0.25	1.95 ± 0.3
	2.68 ± 0.54 ***		1.99 ± 0.29 ***	
BOP (%)	44.39 ± 16.43	43.99 ± 17.29	12.53 ± 4.11	13.64 ± 6.4
	44.16 ± 16.87 ***		13.26 ± 5.70 ***	
BL (proportion of root length)	26.83 ± 8.33	25.27 ± 8.28	15.9 ± 4.36	14.08 ± 4.64
	25.93 ± 8.31 ***		14.83 ± 4.74 ***	
Missing teeth (*N*) BOP (%)	6.28 ± 4.83 **	3.92 ± 4.11**	3.33 ± 3.44	1.98 ± 2.6
	4.92 ± 4.57 ***		2.52 ± 3.03 ***	
**PD Stages**				
Stage I (*N*)	13	20	N.A.	
	33			
Stage II (*N*)	22	24	N.A.	
	46			
Stage III (*N*)	8	15	N.A.	
	23			
Stage IV (*N*)	18	24	N.A.	
	42			
**RA Clinical Parameters and Treatment**				
DAS28-CRP (score)	4.32 ± 1.28	N.A.	3.82 ± 1.66	N.A.
RAID (score)	4.7 ± 2.1	N.A.	3.95 ± 2.45	N.A.
HAQ (score)	0.8 ± 0.59	N.A.	0.86 ± 0.57	N.A.
bDMARDs (*N*)	25 *	N.A.	20 *	N.A.
sDMARDs (*N*)	38	N.A.	23	N.A.
Glucocorticoids (*N*)	40	N.A.	17	N.A.

Abbreviations: bDMARDs: biologic disease-modifying antirheumatic drugs; BL: bone loss; BOP: bleeding on probing; CAL: clinical attachment loss; DAS28-CRP: Disease Activity Score 28-joint count C reactive protein; HAQ: health assessment questionnaire; N.A.: not applicable; PD: periodontitis; PPD: periodontal probing depth; RA: rheumatoid arthritis; RAID: rheumatoid arthritis impact of disease; SD: standard deviation; and sDMARDs: synthetic disease-modifying antirheumatic drugs. Statistical analysis was presented for group comparison (PD+ versus PD−) and for intragroup comparisons (PD + RA+ versus PD + RA−; PD − RA + versus PD−RA−). Significant differences were indicated (*) in pairs with regard to level of statistical significance: * *p* ≤ 0.050, ** *p* ≤ 0.010, and *** *p* ≤ 0.001.

## Data Availability

The datasets analyzed during the current study are available from the corresponding author on reasonable request.

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
