# Peer review of "Gingival Tissue MiRNA Expression Profiling and an Analysis of Periodontitis-Specific Circulating MiRNAs"

_ijms, 2023, doi:10.3390/ijms241511983_

Round 1

Reviewer 1 Report

 Benita Buragaite-Staponkiene et al. Have performed analysis of miRNA in gingival tissues, plasma and saliva of periodontitis patients and analyzed 4 candidate miRNAs in a larger cohort. The study is designed in a stepwise approach and has sizeable patient cohort. There are some comments that should be addressed to better interpret the findings of the study:

1.     The information on targets of these miRNAs or pathways that these associates with is missing. Authors should make an effort to address this aspect.

2.     As authors have pointed out, age is a confounder in the study. Authors should attempt to put information of influence of age on miRNA expression and investigate literature in this aspect. For an association with disease, it is important that age is not a confounder from miRNA perspective, so that altered miRNAs are not because of age being a contributor. If so, then it should be mentioned in the study.

3.     Line 104, Please clarify the design. Were there 4 RA patients in both PD affected tissue and healthy tissues? Is there any information on disease activity and disease duration for the discovery cohort (8 OD affected and 8 healthy tissues)? Were patients having similar disease activity and duration for this cohort?

4.     Line 105: What was the rationale to choose miRNAs in more than 25% of the samples and not 50% for example? Do this means both test and control or 25% in each group?

5.     Was any housekeeping control used for normalization?

6.     Line 141, 220, 230, 232: FC less than 1.5 should not be tested for significance in view of the initial screening based on FC 1.5.

7.     Supplementary figure 2 should be moved to main content and attempted with connected lines to better appreciate the findings.

English language quality is good. 

Reviewer 2 Report

I have reviewed too many papers to count.  This is the first I have read and reviewed that I can see nothing I would change.  The study, methodology, results and interpretation are all top notch.

Reviewer 3 Report

The article is devoted to the study of miRNA associated with periodontitis. Quite interesting associations of periodontitis with the expression of various types of miRNAs have been found. In my opinion, the article is well written, the methods used are adequate to the task, the conclusions are justified. In addition, the high practical significance of this study for medicine should be noted. In principle, the article can be published, but I have a small wish about Figure 1. The fact is that the inscriptions on it are completely unreadable. I would just remove them if possible.  

Reviewer 4 Report

I have reviewed the manuscript entitled “Gingival tissue miRNA expression profiling and analysis of 2 periodontitis-specific circulating miRNAs”. The authors have well identified the need for unearthing miRNAs as predictive disease markers that can help differentiate periodontal health from disease.  The authors are attempting a high throughput approach to reveal the role of miRNAs as disease markers in supporting the development/detection of periodontal disease. Furthermore, the authors pursue that they can leverage the correlation between periodontal clinical characteristics with the presence/absence of the studied biomolecules in gingival tissues, saliva and plasma to explore their ability as diagnostic biomarkers. It is however unclear, if the authors understand or can decipher what it means if levels of certain molecules differ in periodontal disease, given the massive heterogenic changes occurring in the periodontal disease process. My specific comments to this manuscript are as follows:

Introduction: Based on previously published literature, the authors identify the potential functions miRNAs in various disorders such as cancer, immunity, inflammation, etc. However, the manuscript lacks sufficient rationale discussing the role of miRNAs in periodontal disease. A sound justification is warranted as to why miRNAs as biomarkers need to be investigated and may play crucial roles in the pathophysiology of periodontal disease.

What is the specific role of candidate miRNAs in other inflammatory model systems?

Material & Methods: The power analysis is only shown for plasma sample collection from the patients.

Major comment: For the microarray analysis: It is unclear as to why human samples derived from a sample size of only n = 8 each group was chosen. There is no mention of power analysis and neither have the subjects suffering from periodontal disease been clearly defined.

This needs to be re-executed in a larger cohort with a definitive power analysis.

To minimize inter-examiner reproducibility errors, the authors need to document if they employed a single calibrated examiner to evaluate clinical parameters, perform the sample collection.

Results:

Results in Figure 2 and 4 on the Relative expressions of miR-125a-3p, 140-3p, -145-5p, -146a-5p, -195-5p in gingival tissue in Comparison with the mean clinical attachment loss; periodontal probing depth; bleeding on probing; bone loss is a very confusing form of depiction and needs to be re-represented by a chart that is more meaningful and easy to follow what is presented.

Results in Figure 3: MiRNA expression in gingival tissue does not correlate with clinical status of PD. At best there is a weak to moderate correlation with the PD stages. This paragraph and the correlation matrix need to be corrected accordingly.

The authors’ state that on Page 9 that: “in blood plasma a combination of miR-140-3p, miR-145-5p, miR-265 146a-5p and miR-195-5p significant AUC value (AUC = 0.58; 95% CI = 0.51 to 0.66, P = 266 0.036) (Fig 5B).”

Unfortunately, authors’ need to review the AUC value significance as an AUC of 0.58 means the model has no class separation capacity whatsoever.

The non-significant results from Figure 5B need to reword accordingly.

Major comment: The In-vitro validation of the differential expression of the respective miRNAs does not help conclude to the functional impact this might have on the periodontal disease model.

Further in-vitro functional validation experiments with enlisting the potential genes targeted/impacted are warranted to establish a true association of the enlisted miRNAs, in-order to truly gauge the impact of miRNA regulation in periodontitis with / without RA. In-order to truly gauge the impact of miRNA regulation in periodontitis, it is recommended to include in-vitro functional validation with over-expression or knock-down of miRNA and assess for phenotypic changes, if any. Likewise, in-vitro experiments to establish the functional relevance of miRNA candidates is highly recommended (e.g. Cellular localization, immune mediated changes).

Discussion:

This is an over-extrapolation of the presented findings as this does not fall in lines with what is described.

Please discuss the relevance of your results in the context of periodontal disease.

The correlation analysis between the relevant miRNAs only weakly positively correlated and need to be reworded accordingly.

The dynamic response of the periodontal inflammatory status with the miRNA dysregulations is futile unless in-vitro functional verification or K/O strategies with direct periodontal pathogen challenge (simulating in-vitro periodontal inflammation models); are able to validate the functional relevance of this observation. This observation could very well be a secondary outcome of the pathogen induced host immunomodulatory changes.

Additionally, please provide a logical explanation why was a ROC curve analysis for plasma samples can serve as an attractive diagnostic target with only modest to no plausible changes in periodontitis.

Conclusion: While the authors are off to a good start, however, this study requires additional experiments, as stated above. Additionally, the authors should include more information that clarifies and justifies their study rationale and clinical significance, given that most results are of modest significance.

  • I recommend rehashing the manuscript for grammatical errors, typos and phrasing issues.
  • While the study appears to be sound, the language is unclear, making it difficult to follow. An advice to the authors is to work with a writing coach or copyeditor to improve the flow and readability of the text.

Round 2

Reviewer 4 Report

Appreciate the author’s comments to addressing the suggestions requested